# Salicylic Acid Enhances Cadmium Tolerance and Reduces Its Shoot Accumulation in *Fagopyrum tataricum* Seedlings by Promoting Root Cadmium Retention and Mitigating Oxidative Stress

**DOI:** 10.3390/ijms232314746

**Published:** 2022-11-25

**Authors:** Siwei Luo, Kaiyi Wang, Zhiqiang Li, Hanhan Li, Jirong Shao, Xuemei Zhu

**Affiliations:** 1College of Environmental Sciences, Sichuan Agricultural University, Huimin Road No. 211, Chengdu 611130, China; 2College of Life Science, Sichuan Agricultural University, Xinkang Road No. 46, Yaan 625014, China

**Keywords:** Tartary buckwheat, cadmium, salicylic acid, subcellular distribution, chemical forms, transcriptome analysis

## Abstract

Soil cadmium (Cd) contamination seriously reduces the production and product quality of Tartary buckwheat (*Fagopyrum tataricum*), and strategies are urgently needed to mitigate these adverse influences. Herein, we investigated the effect of salicylic acid (SA) on Tartary buckwheat seedlings grown in Cd-contaminated soil in terms of Cd tolerance and accumulation. The results showed that 75–100 µmol L^−1^ SA treatment enhanced the Cd tolerance of Tartary buckwheat, as reflected by the significant increase in plant height and root and shoot biomass, as well as largely mitigated oxidative stress. Moreover, 100 µmol L^−1^ SA considerably reduced the stem and leaf Cd concentration by 60% and 47%, respectively, which is a consequence of increased root biomass and root Cd retention with promoted Cd partitioning into cell wall and immobile chemical forms. Transcriptome analysis also revealed the upregulation of the genes responsible for cell wall biosynthesis and antioxidative activities in roots, especially secondary cell wall synthesis. The present study determines that 100 µmol L^−1^ is the best SA concentration for reducing Cd accumulation and toxicity in Tartary buckwheat and indicates the important role of root in Cd stress in this species.

## 1. Introduction

Cd contamination has become a threat to global food safety due to its high mobility and exchangeable fraction in soils [1]. Cd stress in crops leads to growth inhibition, reduced uptake of essential minerals, decreased chlorophyll content, and increased reactive oxygen species, which eventually results in decreased crop productivity and economic losses [2,3,4]. In 2006, the Ministry of Environmental Protection of China reported that soil contamination led to a 10-million-ton reduction in grain production and high residues of pollutants in 12 million tons of grain, resulting in a direct economic loss of more than CNY 20 billion [5]. Moreover, Cd is readily able to accumulate in the edible parts of crops to levels that are harmful to the diet [6], making dietary intake the main route of cadmium exposure for the nonsmoking population [7]. Therefore, mitigation strategies that can simultaneously promote Cd tolerance and minimize Cd accumulation in the edible parts of crops are urgently needed to ensure the safe production and sustainable development of agriculture.

The application of phytohormones is an efficient and cost-effective measure to achieve the goal of promoting safe crop production in the Cd-contaminated soil, and SA has been widely tested and found to be effective in pursuing the goal in various crops [8,9,10]. SA is an important phytohormone reported to be involved in inducing plant responses to various abiotic stresses, including drought, salt, high or low temperature, and heavy metal stress [11,12,13,14]. Many studies have focused on the application of exogenous SA to improve the abiotic stress tolerance of crops, which is considered to be beneficial to the growth and production of crops under abiotic stress [15,16,17,18,19,20]. It is reported that SA mitigates cadmium toxicity in plants by reducing Cd uptake and root-to-shoot translocation, improving photosynthetic capacity and alleviating oxidative stress [21]. For example, the decreases in plant height, root length, and plant biomass induced by Cd stress were mitigated by 100 µmol L^−1^ SA treatment in rice, with cadmium concentrations in roots and shoots decreasing by 48% and 20%, respectively [22]. Similar results were also observed in oilseed rape, in which 50 µmol L^−1^ SA treatment reduces Cd accumulation in leaves by 20% to 35% [23]. SA treatment increases antioxidant enzyme activities in potato under Cd stress, which results from the dramatic upregulation of gene *StAPX* and *StSOD* [20]. In addition, the genes involved in the synthesis of non-enzymic antioxidants, including glutathione and flavonoids, which confer plant cadmium tolerance, were also found to be induced by SA [24,25,26,27]. SA also plays a role in the induction of synthetic gene expression of lignin and pectin, which are essential components of plant cell walls required for Cd adsorption and retention [22,28,29]. However, the effects of exogenous SA treatment on the cadmium tolerance of plants are dose-dependent and species-specific, and the regulatory mode of the specific signal transduction mechanism of SA in alleviating cadmium toxicity in plants is still unknown.

Tartary buckwheat (*Fagopyrum tataricum*), a multipurpose orphan crop used in the food industry, is mainly cultivated in the alpine and arid regions of China, Russia, Korea, Japan, and India [30,31]. As a functional food raw material with nutritional and medicinal effects, Tartary buckwheat has been processed into noodles, bread, biscuits, tea, and other popular foods [32]. However, it is reported that Tartary buckwheat readily accumulates Cd in its edible parts, resulting in excessive levels of heavy metal elements in some Tartary buckwheat products [33,34,35,36,37]. Although previous studies have shown that Tartary buckwheat responds to heavy metal stress by improving oxidative stress and promoting the compartmentalization of heavy metals in the vacuole [34,36], mitigation strategies capable of enhancing cadmium tolerance and minimizing cadmium accumulation in Tartary buckwheat remain unclear.

Thus, the aim of this research is to (1) assess the feasibility of foliar application of SA to enhance cadmium tolerance and reduce shoot cadmium accumulation in Tartary buckwheat and determine the best concentration for SA application; and (2) investigate the mechanisms of SA affecting Cd tolerance and detoxification in Tartary buckwheat. The results from this study contribute to an increased understanding of the role of SA in Cd tolerance in Tartary buckwheat in addition to the safe production of Tartary buckwheat grown in Cd-contaminated soils.

## 2. Results

### 2.1. SA Promotes Growth and Mitigates Oxidative Stress of Tartary Buckwheat under Cadmium Stress

A soil Cd contamination of 2.0 mg kg^−1^ significantly reduced the shoot length, root length, shoot fresh weight, and root fresh weight of the Tartary buckwheat seedlings by more than 10% (*p* < 0.05), and SA had different effects on these growth parameters at different concentrations (Figure 1). The application of 25 µmol L^−1^ SA further decreased the shoot fresh weight by 6%, but increased the root length by 11% compared to the Cd group. Applications of 50 and 75 µmol L^−1^ SA both increased the shoot fresh weight and root fresh weight. The best performance of the SA treatment in promoting Tartary buckwheat growth was observed at the treatment concentration of 100 µmol L^−1^ (Figure 1). At this application concentration, the shoot length and shoot fresh weight were significantly increased by 11% and 13% (*p* < 0.05), respectively, to the same level of the seedlings of the control group grown in the uncontaminated soil, and the root fresh weight greatly increased by 53% to an even higher level than the seedlings of the control group (Figure 1).

SA treatment at concentrations from 50 to 100 µmol L^−1^ mitigated the oxidative stress caused by Cd contamination, and the performance increased with increasing SA concentration (Table 1). Compared to the seedlings of the control group grown in uncontaminated soils, seedlings grown in the Cd-contaminated soils showed apparent oxidative stress. The hydrogen peroxide (H_2_O_2_) and malondialdehyde (MDA) content increased by 31% and 127%, respectively, and the proline (Pro) content and the activity of peroxidase (POD), superoxide dismutase (SOD), and catalase (CAT) decreased by 36%, 48%, 19%, and 54%, respectively. The 25 µmol L^−1^ SA treatment did not show any obvious effect on the oxidative stress caused by Cd contamination (Table 1). At this concentration, SA largely increased the proline content and the activity of POD, SOD, and CAT, yet only the SOD increase was significant (*p* < 0.05) and the H_2_O_2_ content was increased. However, the oxidative stress was mitigated by SA treatment at concentrations of 50 to 100 µmol L^−1^. A monotonic decrease in the H_2_O_2_ and MDA contents was observed when SA concentrations increased from 50 to 100 µmol L^−1^. The H_2_O_2_ and MDA contents significantly decreased (*p* < 0.05) in response to 75 to 100 µmol L^−1^ SA treatment, and the H_2_O_2_ and MDA contents notably decreased by 39% and 44%, respectively, with 100 µmol L^−1^ SA treatment. The Pro content and the activity of POD, SOD, and CAT also monotonically increased with increasing SA concentration. Treatment with 100 µmol L^−1^ resulted in the Pro content and POD, SOD, and CAT activity greatly increasing by 94%, 51%, 56%, and 191%, respectively, indicating considerable alleviation of oxidative stress at this concentration.

### 2.2. SA Promotes Root Cd Retention and Enhances Cd Partitioning in the Cell Wall and Insoluble Cd Chemical Forms

Cd contamination significantly increased the Cd concentration in all organs of Tartary buckwheat seedlings including the root, stem, and leaf (*p* < 0.05, Figure 2). Under 2.0 mg kg^−1^ soil Cd contamination, the Cd concentration in the root, stem, and leaf increased by 69%, 114%, and 117%, respectively, and the Cd concentrations were higher in the seedling organs than in the soil, corresponding to a bioconcentration factor higher than one. Moreover, it was observed in the experiment that more Cd accumulated in the root than the stem and leaf.

SA treatment altered Cd accumulation in Tartary buckwheat in an organ-dependent manner (Figure 2). The leaf and stem Cd concentration decreased, while the root Cd concentration increased with the increase in SA concentration from 25 to 100 µmol L^−1^. With the 100 µmol L^−1^ SA treatment, the leaf and stem Cd concentrations were considerably reduced by 60% and 47%, respectively (Figure 2A,B), to the same level of the control plant group grown in the uncontaminated soils (*p* < 0.05). However, the root Cd concentration was elevated by 32% with the 100 µmol L^−1^ SA treatment (Figure 2C). The differentiated effect of SA on the Cd accumulation in different organs led to a significant change in the translocation factors (Figure 2D,E). No significant change in the stem-to-leaf translocation factor was observed for treatments with 25 to 75 µmol L^−1^ SA (*p* < 0.05), and the root-to-stem translocation factor decreased when the applied SA concentrations increased from 0 to 100 µmol L^−1^. At 100 µmol L^−1^ SA, the stem-to-leaf translocation factor decreased by 25% and the root-to-stem translocation factor greatly decreased by 59% (Figure 2). These results indicated that SA had a greater effect on Cd accumulation and translocation in the roots than the shoots of Tartary buckwheat under Cd stress and that SA-induced root Cd retention contributed to the enhanced Cd tolerance of this plant.

To further analyze the pathway of salicylic acid-induced Cd accumulation and translocation change in Tartary buckwheat, we investigated the Cd subcellular distribution and chemical forms in Tartary buckwheat roots under cadmium stress with and without SA treatments, and the results are illustrated in Figure 3.

According to the Cd subcellular distribution, the cell wall fraction accumulated most Cd in the root, stem, and leaf, and the proportion of Cd in the cell wall fraction increased with 100 µmol L^−1^ SA treatment (Figure 3A). More than 60% of the Cd in Tartary buckwheat roots was distributed in the cell wall fraction under both SA0 and SA100 treatment (Figure 3A). Nevertheless, SA treatment disturbed the Cd subcellular distribution in Tartary buckwheat organs. The cell wall fraction of the leaf, stem, and root increased by 7%, 8%, and 3% (Figure 3A), respectively, implying that SA treatment blocks Cd accumulation in protoplasts by increasing Cd adsorption on the cell wall, thereby alleviating the phytotoxicity of Cd to Tartary buckwheat and weakening its Cd translocation ability.

In addition, the mobile Cd chemical forms decreased and the less-mobile Cd chemical forms increased in the Tartary buckwheat under SA treatment (Figure 3B). SA treatment reduced the proportion of more toxic and mobile soluble Cd, extracted using ethanol and water, by 7%, 2%, and 7% in the root, stem, and leaf, respectively, and increased the proportion of less toxic and immobile insoluble forms of phosphate complexes and Cd oxalate, extracted using HAc and HCl, by 7%, 6%, and 7% in the root, stem, and leaf, respectively (Figure 3B). Thus, the improved Cd tolerance of Tartary buckwheat with SA treatment can also be attributed to the reduced Cd partitioning in the toxic mobile forms, and Cd translocation could also be limited under SA treatment due to the increased partitioning of Cd into immobile chemical forms.

### 2.3. mRNA Profiling Reveals Upregulation of Genes Involved in Secondary Cell Wall Synthesis and Oxidative Stress Response in the Root

mRNA sequencing was used to compare roots of Tartary buckwheat following either 100 µmol L^−1^ SA treatment or without SA treatment to reveal genes that are differentially expressed between these conditions. In the mRNA sequencing of the six root samples, 6.5 to 7.6 billion clean reads were documented, with the error rate ranging from 2.6% to 2.8% and Q30 percentage ranging from 91% to 93%. A total of 4.3 to 5.0 billion reads were mapped using the reference genome of Tartary buckwheat, and 100% of the total transcripts were annotated using NR, GO, KEGG, EggNOG, and Swiss-Prot databases, with 13k (38%) to 30k (91%) transcripts annotated per database. The transcriptome analysis revealed 29,507 expressed genes in total (Appendix A), in which 26,918 genes were expressed in both groups, 1156 genes were specifically expressed in the Cd group, and 1426 genes were specifically expressed in the SA treatment group (Figure 4A). Using R package DESeq2, 1151 genes were identified as differentially expressed genes with an adjusted *p*-value < 0.10 and absolute log_2_ fold-change > 0.05, including 826 upregulated genes and 325 downregulated genes (Figure 4B,C). In the heatmap of all 1151 DEGs, all of the SA-treated replicates clustered together and all of the Cd group replicates clustered together, indicating a consistent gene expression pattern within each of the two sample groups and between the groups (Figure 4D).

Enrichment analysis using gene ontology (GO) and KEGG ontology (KO) terms was applied to investigate the function of the DEGs as a whole and reveal the biological processes or pathways regulated by SA in Tartary buckwheat root. Six of the top ten enriched GO terms are in categories related to the synthesis and metabolism of cell wall components and the cell wall, including the carbohydrate metabolic process and catabolic process-containing polysaccharide catabolic process, phenylpropanoid catabolic process, lignin catabolic process, and pectin catabolic process categories. Two of the top ten enriched GO terms belong to plant autophagy, and another two GO terms belong to transcription factors (Figure 5A). Among the top ten enriched GO terms, phenylpropanoid catabolic process and lignin catabolic process had the highest Rich factor of above 0.3, followed by the Rich factors of about 0.1 for the pectin catabolic process and polysaccharide catabolic process (Figure 5C), indicating that SA extensively regulates the genes responsible for cell wall biosynthesis in the roots of Tartary buckwheat. According to the distribution of the expression level of the genes and their associated GO terms, the genes associated with the six terms related to cell wall biosynthesis were mostly upregulated, the genes related to autophagy were all downregulated, and those corresponding to the term transcription factor had a mixed expression pattern, with both upregulated and downregulated genes (Figure 5E). The GO term enrichment analysis above indicates that the increased root Cd retention and cell wall Cd could be explained by the upregulation of the genes in the cell wall biosynthesis.

Among the top ten enriched KO terms, three terms were also related to the biosynthesis of the cell wall, including phenylpropanoid biosynthesis, glycosyltransferases, and stilbenoid, diarylheptanoid, and gingerol biosynthesis (Figure 5B). Other enriched KO terms include the plant MAPK signaling pathway, plant hormone signal transduction, cytochrome P450, biosynthesis of various plant secondary metabolites, cysteine and methionine metabolism, zeatin biosynthesis, and plant–pathogen interaction (Figure 5B). The highest Rich factor of 0.18 was found for zeatin biosynthesis, followed by stilbenoid, diarylheptanoid, and gingerol biosynthesis with 0.14, and then phenylpropanoid biosynthesis, cytochrome P450, and biosynthesis of various plant secondary metabolites with about 0.1 (Figure 5D). The ridge plot of the gene expression level of the KO terms shows that the genes of six terms were generally upregulated, including three terms related to cell wall synthesis and another three terms including the biosynthesis of various plant secondary metabolites, cysteine and methionine metabolism, and plant–pathogen interaction, which could have played a role in Cd detoxification and oxidative stress mitigation (Figure 5F). The genes of another four terms possessed a mixed expression pattern with approximately even up- and downregulation. The results above reveal that SA induces extensive regulation of the genes responsible for different pathways, with emphasis on the pathways related to cell wall biosynthesis.

The DEGs of the top ten enriched GO terms with absolute log_2_ fold-change >2 are listed in Figure 6, and their annotations were analyzed in detail. More than 50% of the genes with known functions are involved in cell wall metabolism, and most were induced by SA treatment. The genes of laccase 2 (*FtPinG0201546600.01*), laccase 4 (*FtPinG0302685700.01*), laccase 12 (*FtPinG0100282900.01*), and laccase 14 (*FtPinG0100173000.01*, *FtPinG0100173200.01*, and *FtPinG0100173400.01*) were upregulated, indicating enhanced lignin biosynthesis in the secondary cell wall. In addition, the following DEGs also participate in secondary cell wall biosynthesis and were upregulated by SA treatment: the genes of probable beta-D-xylosidase 2 (*FtPinG0100711600.01*) and probable beta-1,4-xylosyltransferase (*FtPinG0505214600.01*), which act on xylan biosynthesis, remodeling, and deposition. Cellulose synthase A catalytic subunit 7 (*FtPinG0808212600.01*) affects secondary cell wall deposition, cationic peroxidase 1 (*FtPinG0101136800.01*) functions in the synthesis and degradation of lignin, and NAC domain-containing protein 30 (*FtPinG0505150700.01*) acts as a transcription activator of the genes involved in the secondary cell wall synthesis and xylan accumulation, which could have induced the expression of the abovementioned genes.

The expression of many genes that predominantly play a role in primary cell wall metabolism was also induced by SA treatment, including FtPinG0303251600.01 and *FtPinG0380000816.01* coding probable pectate lyase 12, *FtPinG0100989200.01* coding probable pectate lyase 22, and *FtPinG0505573300.01* and *FtPinG0808584200.01* coding probable pectate lyase 8 (Figure 6). Pectate lyases are linked with cell expansion and their induction by SA treatment could have contributed to a faster root growth. The upregulation of *FtPinG0302532900.01* (SD16), *FtPinG0100742600.01* (SKP1A), *FtPinG0101027900.01* (CEL1), and *FtPinG0505150700.01* (NAC030) also support faster cell expansion and division (Figure 6), which could contribute to increased root fresh weight. However, another important DEG responsible for cell expansion and elongation, *FtPinG0808411100.01* (XTH22), was found to be downregulated (Figure 6). In addition to cell wall metabolism, the mRNA sequencing also revealed that the expression of peroxidase genes *FtPinG0101136800.01* (PER1), *FtPinG0606354600.01* (PER1), *FtPinG0606123000.01* (PER3), *FtPinG0202424900.01* (PER4), *FtPinG0100423800.01* (PER50), and *FtPinG0707543500.01* (PER57) was also upregulated by SA treatment (Figure 6). These genes play essential roles in the removal of H_2_O_2_ as well as lignin synthesis. Their upregulation corresponds to both the enhancement of cell wall synthesis and the alleviation of oxidative stress in the SA-treated Tartary buckwheat observed in the experiment.

The genes corresponding to the top ten KO terms resulting from KEGG enrichment analysis differed to those of GO enrichment analysis (Figure 7). Apart from the overlapping genes, such as the genes coding cellulose synthase, cationic peroxidase 1, and peroxidase 3, 4, 50, and 57, the KEGG enrichment analysis revealed more genes responsible for plant hormone signal transduction. The expression of gene *FtPinG0505861900.01*, which codes abscisic acid (ABA) negative regulator abscisic acid 8′-hydroxylase, was suppressed (Figure 7), indicating the potential positive regulation of ABA-induced stress responses. Other hormone-signaling related genes were upregulated with SA treatment, including the genes for abscisic acid receptor PYL4 (*FtPinG0505812800.01*), auxin-responsive protein IAA26 (*FtPinG0404298400.01*), auxin transporter-like protein 2 (*FtPinG0505007000.01*), repressor of jasmonate responses protein TIFY 6B (*FtPinG0404161200.01*), and cytochrome P450 94C1, which is involved in the jasmonate-mediated signaling pathway (*FtPinG0606733300.01*). Two obviously upregulated genes, i.e., the hormone signal transduction genes *FtPinG0202059400.01* (PR1) and *FtPinG0303395300.01* (PRB1), are involved in plant–pathogen interaction and may also contribute to alleviating oxidative stress in Tartary buckwheat. The upregulation of genes of calmodulin *FtPinG0708110100.01* (CALM) and *FtPinG0302843600.01* (CAM-1) could induce the plant tolerance to a series stresses including salt, metal, osmotic, and oxidative stress. The induction of the genes in the MAPK signaling pathway indicates the role of hormone signaling transduction resulting from SA treatment. The regulation of signaling genes also indicates the presence of a complex interaction network between applied exogenous SA and various endogenous hormones in Tartary buckwheat.

## 3. Discussion

In the present study, SA treatment promoted the growth of Tartary buckwheat seedlings in Cd-contaminated soil. Increases in plant height and fresh weight under SA treatment have been observed in numerous studies in a wide variety of plant species grown in the presence of various heavy metals [21,38,39,40,41,42,43,44]. Recent studies on rice, wheat, and mung bean have also reported the capability of exogenous SA to alleviate the root length inhibition caused by Cd stress [40,41,42,43,44]. However, our experiment shows that 50–100 µmol L^−1^ SA had no significant influence on the root length of Tartary buckwheat, which could result from the inhibition effect of SA on the root elongation. For example, it was reported in rice that exogenous SA inhibits root elongation under aluminum stress [45]. These results demonstrate a complex effect of SA on root length, which is also indicated by the contradictory effect of endogenous SA and the dose-dependent exogenous effect without heavy metal stress [46]. Nevertheless, in this study, the root fresh weight greatly increased with SA treatment. The contrary effect of SA on root length and root weight could be a result of the induction of *FtPinG0100742600.01* (SKP1A) and *FtPinG0505150700.01* (NAC030). The upregulation of SKP1 reduces root length in *Arabidopsis thaliana* [47], and the overexpression of the soybean NAC gene *GmNAC109* significantly increases the number of lateral roots [48]. Therefore, the induction of these two genes could lead to increased root biomass with shorter root length in the SA-treated Tartary buckwheat.

Another growth-promoting factor is oxidative stress mitigation caused by high Cd concentration. In the reduction of molecular oxygen by heavy metals, reactive oxygen species (ROS) are produced, which leads to the oxidative decomposition of membrane lipids and proteins [49]. The decreased H_2_O_2_ and MDA concentration indicate the alleviation of oxidative stress in the SA-treated Tartary buckwheat, and the increases in proline content and SOD, POD, and CAT activities indicate higher ROS-scavenging capacity, which is consistent with the findings of other studies using different plant materials [10,39,50]. Furthermore, multiple class III peroxidase genes were upregulated by SA treatment, which supports H_2_O_2_ detoxification [51]. These peroxidases are also involved in the repair of the plasma membrane, whose permeability is affected by lipid peroxidation under oxidative stress [51].

It should also be emphasized that growth was promoted and stress was mitigated due to decreased Cd content resulting from SA treatment. In our study, the Cd concentrations in the stem and leaf were materially reduced with 100 μmol L^−1^ SA treatment to a level that was no significantly different to that of the plant group grown in the uncontaminated soil. The decrease in Cd accumulation under SA treatment has also been observed in other crops [22,41,52]. By contrast, we found that the root Cd concentrations greatly increased in response to SA treatment in our study. In rice plants under 5 μmol L^−1^ Cd stress, the root Cd concentrations decreased by 48% with 100 μmol L^−1^ SA treatment [22]. For SA-treated pea seedlings hydroponically grown with Cd concentrations of 1 and 2 μM, the root Cd concentrations decreased by 61% and 72%, respectively [53]. A more moderate, but significant, decrease in root Cd with SA treatment has also been observed in wheat, maize, and mung bean under Cd stress [41,44,52]. The opposite trends of changes in Cd concentration in roots and shoots are consistent with a large reduction in Cd mobility under SA treatment. In our experiment, support for reduced Cd mobility was obtained by the observation of subcellular distribution and chemical form distribution, with increased Cd partitioning in the cell wall, and in the immobile chemical forms of phosphate complex and Cd oxalate [54].

The opposite change in Cd accumulation in the root and stem of SA-treated Tartary buckwheat led to considerably decreased Cd root-to-shoot translocation. Decreased root-to-shoot translocation has also been observed for rice, maize, and ryegrass, but not mung bean, which indicates a species-specific response of Cd root-to-shoot translocation to SA [22,41,52,55,56]. Notably, according to the transcriptome analysis, genes with the biofunction of metal transporters were not enriched, as reported in many other studies [57,58], but, rather, those that were predominantly enriched function in cell wall synthesis, especially in the synthesis and deposition of secondary cell wall components. The upregulation of laccase genes, indicating enhanced lignin crosslinking, may contribute to increased Cd tolerance and root-to-shoot partitioning [59,60]. In addition, SA-inducing lignin formation was observed in common buckwheat and wheat under Cd stress, which provides evidence for the relationship between lignin formation and the cadmium stress response [61]. The upregulation of peroxidases is also reported to be crucial for lignin deposition in roots [62]. In addition, the cellulose synthase A catalytic subunit 7 coded by *FtPinG0808212600.01* forms a cellulose synthase complex for the cellulose synthesis of the secondary cell wall, and probable beta-1,4-xylosyltransferase IRX14 coded by *FtPinG0505214600.01* is involved in the synthesis of the hemicellulose glucuronoxylan to achieve the successive addition of xylosyl residues during xylan backbone elongation [63]. These DEGs could be directly activated by NAC domain-containing protein 30 [64], a transcriptional activator, which was also obviously upregulated by SA treatment in this experiment. These findings are consistent with the studies on the antipathogenic role of SA, in which SA treatment enhances the synthesis of secondary cell wall [65], and the enhancement contributes to the special Cd retention ability of the Tartary buckwheat roots under SA treatment in our study.

Although SA increased root Cd concentration in Tartary buckwheat, the promotion of growth and reduction of stem and leaf Cd accumulation indicate its great potential in the safe production of Tartary buckwheat in Cd-contaminated soils. SA has been widely proved effective in the improvement of plant biomass and fruit yield in various species when the plants are grown under stress conditions such as drought stress, water stress, salinity stress, and heavy metal stresses including Cd stress [29,63,66,67,68,69,70,71,72,73]. In terms of Cd stress, the enhancement of plant growth and alleviation of oxidative stress is believed to contribute to the yield increase with SA treatment [70], which was also observed in our study. In addition, SA treatment reduced the Cd translocation and accumulation in shoots, contributing to the limitation of Cd in edible parts. Tartary buckwheat is very tolerant to high Cd concentration [34], which makes Cd risk more unnoticeable in this crop. Therefore, limiting Cd accumulation in shoots is of particular importance. Additionally, due to the huge costs and long time required by soil remediation, the phytoexclusion of Cd using this highly tolerant crop contributes to the utilization of slightly to moderately contaminated soils [74,75,76]. According to our findings, we suggest applying 100 µmol L^−1^ SA for better and safer production of Tartary buckwheat in Cd-contaminated soils, and further studies on the role of the root cell wall in the SA-induced Cd tolerance in Tartary buckwheat should be promising.

## 4. Materials and Methods

### 4.1. Soil Preparation

The soil was sampled from the plaggic horizon of an agricultural field located at 104°29′ E, 31°39′ N in Sichuan Province, China [77]. The soil is a hydragric anthrosol with pH_water_ of 7.0 and a silt loam texture. The soil organic matter content of was 35 g kg^−1^ and Cd concentration was 0.31 mg kg^−1^. Roots and rocks in the soil were removed, and the soil was air dried at room temperature, ground, and then sieved through a 2 mm sieve. A total of 3 kg soil was used to filled pots with a diameter of 21 cm and height of 20 cm. For the SA treatment groups and Cd group, the filled soil was irrigated with CdCl_2_·2.5H_2_O solution to adjust the Cd concentration to 2.0 mg kg^−1^ [78]. At this concentration, agronomic measures to reduce the risk of Cd contamination are required according to the soil environment quality risk control standard for the soil contamination of agriculture land GB15618-2018. For the control group with uncontaminated soil, deionized water was applied in the irrigation instead. The leachate was subjected to repeated irrigation for a week, and the soil in pots was then incubated for a month. The soil was regularly mixed during the incubation.

### 4.2. Pot Experiment of Tartary Buckwheat

Tartary buckwheat of the cultivar common in southwestern China, Chuanqiao No.1, was grown in pots from March 2021 to April 2021 in the greenhouse of Sichuan Agricultural University, Chengdu, China. The seeds were disinfected with 10% NaClO solution for 20 min and soaked with deionized water for two hours. Then, 15 seeds were sown in each pot, and five seedlings were kept for each pot after 10 days.

According to the comparable studies in other crops, 25–100 µmol L^−1^ salicylic acid water solutions were applied to Tartary buckwheat seedlings [9,22,23]. The application started from the 20th day after sowing. The solution was sprayed manually on the leaves five times at 18:00 every two days until a drop of solution fell from the leaves. The pots were randomly rearranged during the experiment. For the group without SA treatment, deionized water was applied instead of SA solution. Each treatment group comprised three replicate pots.

The seedlings were harvested three days after the final SA application. The root samples were washed with tap water, soaked with 20 mM EDTA for 30 min, and rinsed with deionized water to remove adsorbed Cd. The height, root length, root fresh weight, and shoot fresh weight were measured, and the root, stem, and leaves were sampled for chemical and biochemical analysis.

### 4.3. Measurement of Cd in Organ Level and Subcellular Fractions

Parts of the roots, stems, and leaves were dried at 105 °C for 30 min and then kept at 70 °C for 24 h. The dried samples were ground to powder. Then, 0.2 g powder was sampled and digested in 10 mL mixed acid solution with 4:1 HNO_3_–HClO_4_ (*v*/*v*). The digested solution was filtered through a 0.45 µm membrane. The national standard samples GBW07603 and GBW10019 were also digested using the same procedure. After digestion and filtration, the Cd content of the samples was measured using ICP–MS (NexION 300 ICP–MS spectrometer). The recovery rate ranged from 92% to 103%, supporting the reliability of the measurement.

The cell wall, cell organelle, and soluble fraction of the root, stem, and leaf samples was separated using Weigel’s approach [79]. The frozen fresh samples were ground shortly after sampling with a medium of 0.25 M sucrose, 50 mM Tris–HCl (pH 7.5), and 1 mM dithioerythritol at 4 °C. Then, the mixture was centrifuged at 300× *g* for 30 s, and the pellet was sampled for the cell wall fraction. Subsequently, the supernatant was centrifuged at 20,000× *g* for 45 min to precipitate the organelle fraction. The supernatant collected after the second centrifugation is referred to as the soluble fraction. Finally, the resultant subcellular fractions of the root, stem, and leaf samples were dried, digested, and measured using the same approach for organ Cd measurement.

### 4.4. Measurement of Cd Chemical Form Distribution

A total of 2 g of the frozen root, stem, and leaf samples was ground and mixed with 37.5 mL 80% ethanol [54]. The mixture was shaken at 30 °C for 18 h and then centrifuged at 5000× *g* for 10 min. The supernatant was collected, and the pellet was resuspended [54]. The procedure was repeated three times, and the collected supernatant was the fraction of inorganic soluble Cd, including nitrate/nitrite, chloride, and aminophenol Cd [54]. A series of the same extraction procedure was applied one by one using the following extraction solutions: distilled water for organic soluble Cd in organic acid and Cd(H_2_PO_4_)_2_ forms, 1 M NaCl for Cd integrated with pectates and protein, 2% acetic acid for insoluble CdHPO_4_, Cd_3_(PO_4_)_2_ and other Cd phosphate complexes, and 0.6 M HCl for Cd oxalate [54]. The unextracted remains comprise the residual form of Cd. The collected extracts were dried, digested, and measured using the same approach as described in Section 4.3.

### 4.5. Measurement of H_2_O_2_, MDA, and Proline Content

H_2_O_2_ was measured following the approach described by Hossain [80]. About 0.5 g of fresh leaf sample was homogenized in 50 mM K-phosphate buffer (pH = 6.5) at 4 °C. The mixture was centrifuged at 11,500× *g* for 15 min. Then, 3 mL of supernatant was mixed with 1 mL 0.1% TiCl_4_ in 20% H_2_SO_4_ (*v*/*v*), and the mixture was centrifuged at 11,500× *g* for 15 min. H_2_O_2_ was then measured at 410 nm via absorption spectrometry (UV–Vis Spectrophotometer L6, Shanghai Yoke Instrument Co., Ltd., Shanghai, China).

MDA was measured using trichloroacetic acid as reagent as described by Hossain [81]. A 0.5 g leaf sample was homogenized in 3 mL of 5% (*w*/*v*) trichloroacetic acid and centrifuged at 11,500× *g* for 10 min. Then, 1 mL supernatant was mixed with 4 mL TBA reagent prepared as 0.5% of TBA in 20% TCA. The mixture was heated at 95 °C for 30 min, cooled, and then centrifuged at 11,500× *g* for 15 min. The absorbance was measured at 532 nm with correction using the absorbance at 600 nm. The concentration of MDA was calculated using the extinction coefficient of 155 mM^−1^ cm^−1^.

Proline was measured using the approach described by Trotel [82]. The water-soluble compounds were extracted by heating the leaf sample in deionized water in a boiling water bath. Ninhydrin reagent was prepared by dissolving 0.5 g of ninhydrin in 30 mL of glacial acetic acid and 20 mL of water. Then, a 150 μL aliquot of the extracted sample was mixed with 1 mL ninhydrin reagent and incubated in a boiling water bath for 20 min. The mixture was cooled in an ice-water bath, and the formed product was extracted using 3 mL of toluene. The resultant solution was measured at 520 nm using absorption spectrometry.

### 4.6. SOD, POD, and CAT Enzyme Activity Assay

A 0.5 g leaf sample was homogenized with a pre-cooled mortar and pestle in 1 mL 50 mM ice-cold K-phosphate buffer (pH 7.0) at 4 °C, which contains 100 mM Kill, 1 mM ascorbate, 5 mM β-mercaptoethanol, and 10% (*w*/*v*) glycerol. Then, the resulting mixture was centrifuged at 11,500× *g* for 10 min at 4 °C [80]. The SOD, POD, and CAT activities in the resulting supernatant were assayed by strictly following the approach described by Hossain [80].

### 4.7. Transcriptome Analysis of Root

Total RNA was extracted from the tissue using Plant RNA Purification Reagent for plant tissue according to the manufacturer’s instructions (Invitrogen), and genomic DNA was degraded using DNase I (TaKara). RNA degradation and contamination was monitored on 1% agarose gels. Then, the quality of the RNA was determined using a 2100 Bioanalyser (Agilent Technologies), and it was quantified using the ND-2000 (NanoDrop Technologies). Only high-quality RNA samples (OD260/280 = 1.8~2.2, OD260/230 ≥ 2.0, RIN ≥ 8.0, 28S:18S ≥ 1.0, >1 μg) were used to construct the sequencing library.

RNA purification, reverse transcription, library construction, and sequencing were performed at Shanghai Majorbio Bio-pharm Biotechnology Co., Ltd. (Shanghai, China) and according to the protocols supplied by the instrument manufacturer (Illumina, San Diego, CA, USA). The transcriptome library was prepared using the TruSeqTM RNA sample preparation kit from Illumina (San Diego, CA, USA) as recommended, and using 1 μg of total RNA. Firstly, messenger RNA was isolated according to the polyA selection method with oligo(dT) beads and then fragmented in fragmentation buffer. Secondly, double-stranded cDNA was synthesized using a SuperScript double-stranded cDNA synthesis kit (Invitrogen, CA, USA) with random hexamer primers (Illumina). Then, the synthesized cDNA was subjected to end-repair, phosphorylation, and ‘A’ base addition according to Illumina’s library construction protocol. The libraries were size-selected for cDNA target fragments of 300 bp on 2% low-range ultra agarose followed by PCR amplification using Phusion DNA polymerase (NEB) for 15 PCR cycles.

After being quantified by TBS380, the paired-end RNA-seq sequencing library was sequenced with the Illumina NovaSeq 6000 sequencer (2 × 150 bp read length). The raw paired-end reads were trimmed and quality controlled by fastp (https://github.com/OpenGene/fastp (accessed on 15 February 2022)) [83] with default parameters. Then, clean reads were separately aligned to the reference genome of *Fagopyrum tataricum* [84] with orientation mode using HISAT2 (http://ccb.jhu.edu/software/hisat2/index.shtml (accessed on 15 February 2022)) software [85]. The mapped reads of each sample were assembled by StringTie (https://ccb.jhu.edu/software/stringtie/ (accessed on 15 February 2022)) in a reference-based approach [86].

The expression levels were calculated using the reads counts of the clean reads, and transcripts per million (TPM) values were calculated to normalize the total expression level of the sample for comparison [87]. The function of the genes was annotated using BLAST+ program version 2.9.0 with NR, Swiss-Prot, EggNOG, GO, and KEGG databases [88]. The DEGs were identified using R package DESeq2 version 1.36.0 [89], applying adjusted *p*-value < 0.1 and absolute log2 fold-change > 0.5 to control the false discovery rate and true positive rate [90]. Then, the enrichment analysis of the GO and KO terms of the DEGs was performed using the functions “gseGO” and “enricher” of the R package “clusterProfiler” version 4.4.4 [91].

### 4.8. Statistical Analysis and Data Visualization

The calculation, statistical tests and data visualization were carried out in R software version 4.1.0 [92]. The normality of the duplicates was tested and confirmed using Shapiro–Wilk *t*-tests. Then, a one-way ANOVA model was built using R function “aov”, and the Tukey honest significant differences test was performed by “TukeyHSD” to compare the mean value of the groups [92]. The line plots, dot plots, and bar charts were drawn using R package “ggplot2” version 3.3.6 [93]. For the visualization of gene expression data, Venn diagrams were drawn using R package “ggvenn” version 0.1.9 [94], volcano plots were drawn using R package “EnhancedVolcano” version 1.14.0 [95], emap plots were drawn using R package “enrichplot” version 1.16.2 [96], and ridge plots were drawn using R package “ggplot2” and “ggridges” version 0.5.4 [97]. The heatmaps of DEG expression with hierarchical clustering were created using the R package “pheatmap” version 1.0.12 [98]. Z-scores were calculated to normalize the expression levels of the DEGs for the heatmap, and complete linkage was applied in the hierarchical clustering of the heatmap [98].

## 5. Conclusions

In the present study, we found that application of exogenous salicylic acid can enhance the Cd tolerance of Tartary buckwheat and reduce the Cd accumulation in the shoot of the plant, and 100 µmol L^−1^ SA had the best performance in this crop. SA increased the growth of Tartary buckwheat and mitigated the oxidative stress caused by Cd stress. The enhanced Cd tolerance was also due to decreased Cd accumulation in the stem and leaf with SA treatment. The root of Tartary buckwheat accumulated more Cd with SA treatment and largely reduced the upward Cd translocation. In addition, the Cd translocation from stem to leaf was reduced. The enhanced Cd retention ability could be explained by increased Cd partitioning in the cell wall fraction and immobile chemical forms of Cd phosphate complex and Cd oxalate. Moreover, the transcriptome analysis of the root revealed upregulations of the genes responsible for the cell wall biosynthesis, especially secondary cell wall synthesis with SA treatment. The upregulation could contribute to the largely increased root biomass, root Cd accumulation, and Cd retention. According to our study, the promotion of Cd tolerance and inhibition of shoot Cd accumulation with SA treatment contribute to both the production and product quality of Tartary buckwheat, and further studies on the role of the root cell wall in the SA-induced Cd tolerance in this species are of particular interest.

## Figures and Tables

**Figure 1 ijms-23-14746-f001:**
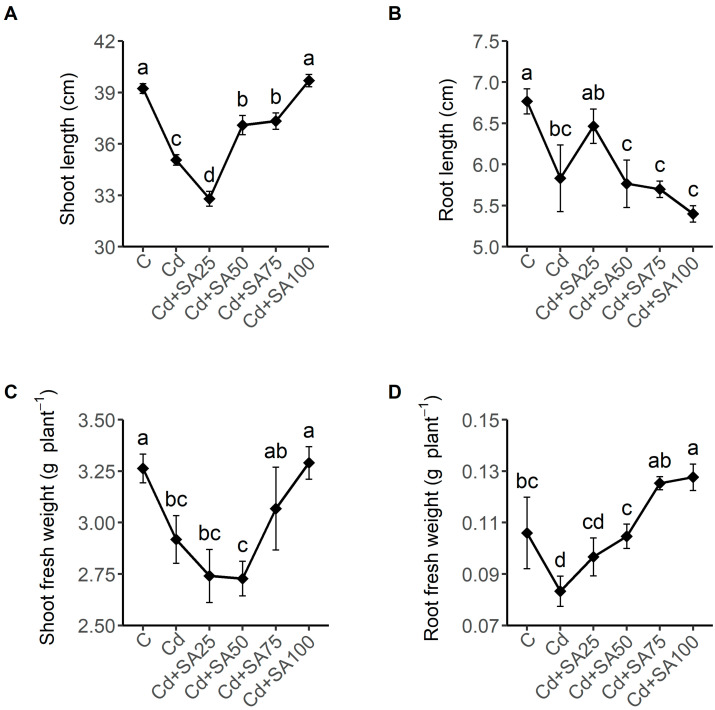
Effect of salicylic acid of different concentrations on the shoot length (**A**), root length (**B**), shoot fresh weight (**C**), and root fresh weight (**D**) of Tartary buckwheat cultivated in soil with Cd contamination. C denotes the control plant group without any treatment cultivated in the uncontaminated soil. Cd denotes the group of plants cultivated in the soil with 2.0 mg kg^−1^ Cd contamination without SA treatment. Cd + SA25, Cd + SA50, Cd + SA75, and Cd + SA100 denote the plant groups cultivated in the soil with 2.0 mg kg^−1^ Cd contamination treated with 25, 50, 75, and 100 µmol L^−1^ salicylic acid, respectively. Different lowercase letters denote significant differences between the treatment groups (*p* < 0.05).

**Figure 2 ijms-23-14746-f002:**
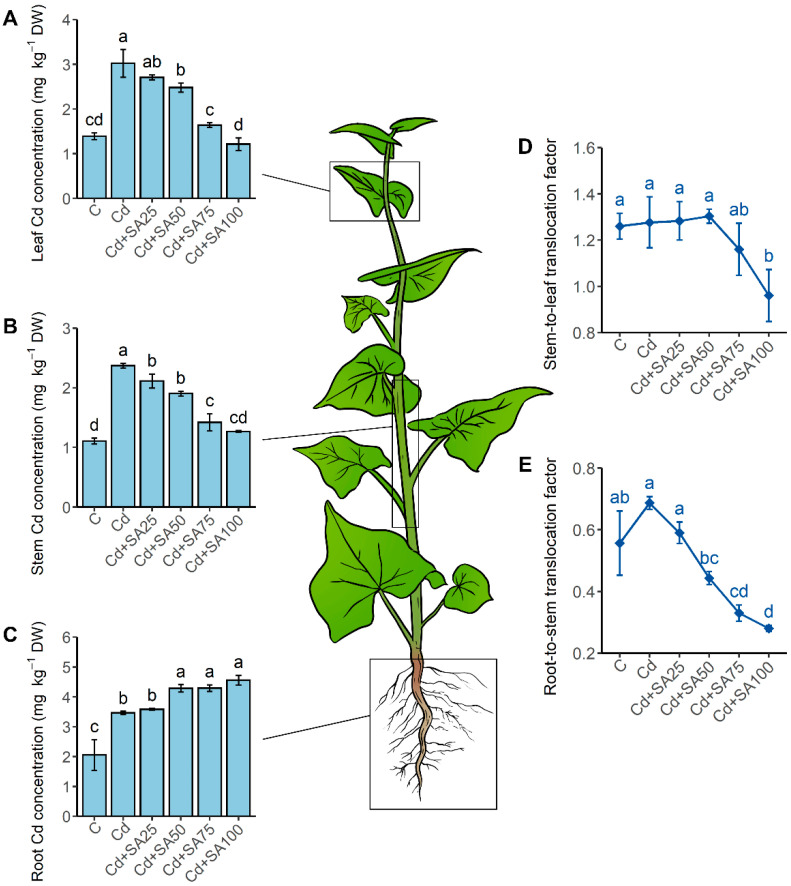
Cd concentrations in the leaf (**A**), stem (**B**), and root (**C**) of Tartary buckwheat cultivated in Cd-contaminated soil treated with different concentrations of salicylic acid and the corresponding stem-to-leaf (**D**) and root-to-stem (**E**) translocation factor. The Cd concentration was measured in mg Cd per kg dry weight of the plant sample (mg kg^−1^ DW). The root-to-stem translocation factor is the ratio of stem Cd concentration to root Cd concentration, and the stem-to-leaf concentration is the ratio of leaf Cd concentration to stem Cd concentration. C denotes the control plant group cultivated in the uncontaminated soil without any treatment. Cd denotes the group of plants cultivated in the soil with 2.0 mg kg^−1^ Cd contamination without SA treatment. Cd + SA25, Cd + SA50, Cd + SA75, and Cd + SA100 denote the plant groups cultivated in the soil with 2.0 mg kg^−1^ Cd contamination treated with 25, 50, 75, and 100 µmol L^−1^ salicylic acid, respectively. Different lowercase letters denote significant differences between the treatment groups (*p* < 0.05).

**Figure 3 ijms-23-14746-f003:**
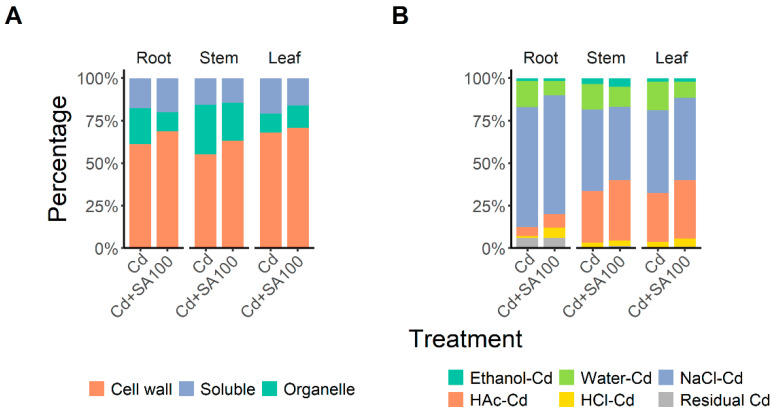
Effects of salicylic acid on the subcellular distribution and chemical form distribution of Cd in the root, stem, and leaf of Tartary buckwheat grown in Cd-contaminated soil. (**A**) Distribution of Cd in the cell wall, organelle, and soluble fractions. (**B**) Chemical form distribution of: ethanol-Cd: inorganic water-soluble Cd extracted using 80% ethanol; Water-Cd: organic water-soluble Cd extracted using deionized water; NaCl-Cd: pectate and protein-integrated Cd extracted using 1 M NaCl; HAc-Cd: insoluble Cd in the form of phosphate complexes extracted using 2% acetic acid water solution; HCl-Cd: insoluble Cd as Cd oxalate extracted using 0.6 M HCl water solution; residual Cd: the final sediment in the extraction procedure. “Cd” denotes the plant group cultivated in the soil with 2.0 mg kg^−1^ Cd contamination without salicylic acid treatment; “Cd + SA100” denotes the plant group cultivated in the soil with 2.0 mg kg^−1^ Cd contamination with 100 µmol L^−1^ salicylic acid treatment.

**Figure 4 ijms-23-14746-f004:**
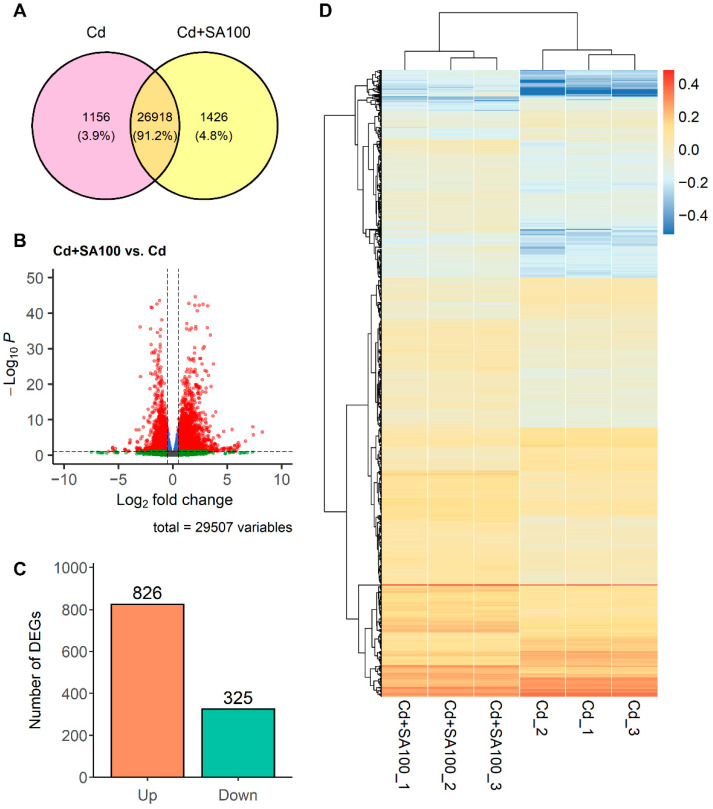
Differential gene expression in the root of Tartary buckwheat plants cultivated in soil with 2.0 mg kg^−1^ Cd contamination with and without 100 µmol L^−1^ salicylic acid treatment. Differential gene expression analysis was applied using R package “DESeq2”. A default significance cutoff of 0.1 was applied in the analysis and the cutoff for low counts was four. A threshold of absolute log_2_ fold-change greater than 0.5 was applied to increase the reliability of the analysis with a sample size of three. (**A**) Venn diagram of the expressed genes between the Cd (plant group without salicylic acid treatment) and the SA100 (plant group with 100 µmol L^−1^ salicylic acid treatment); the overlapping region shows the shared gene expressed between the two groups, and the non-overlapping region is genes exclusively detected for each group. (**B**) Volcano plot of the genes. The plot was created using the R package “EnhancedVolcano”. Each point represents a gene, and the red points denote the differentially expressed tested genes with adjusted *p*-value > 0.1 and absolute value log_2_ fold-change larger than 0.5. The horizontal dash line indicates the adjusted *p*-value of 0.1, and the vertical dash lines indicates the log_2_ fold-change of −0.5 and 0.5. (**C**) Number of differentially expressed genes categorized as either upregulated (Up) or downregulated (Down) genes. (**D**) Heatmap of all differentially expressed genes with hierarchical clustering. SA100_1, SA100_2, and SA100_3 each denote one of the three plant replicates from the plant group cultivated in the soil with 2.0 mg kg^−1^ Cd contamination with 100 µmol L^−1^ salicylic acid treatment. Cd_1, Cd_2, and Cd_3 each denote one of the three plant replicates from the Cd group cultivated in the soil with 2.0 mg kg^−1^ Cd contamination without salicylic acid treatment. The color indicates the expression level in z-score of the TPM values of all differentially expressed genes.

**Figure 5 ijms-23-14746-f005:**
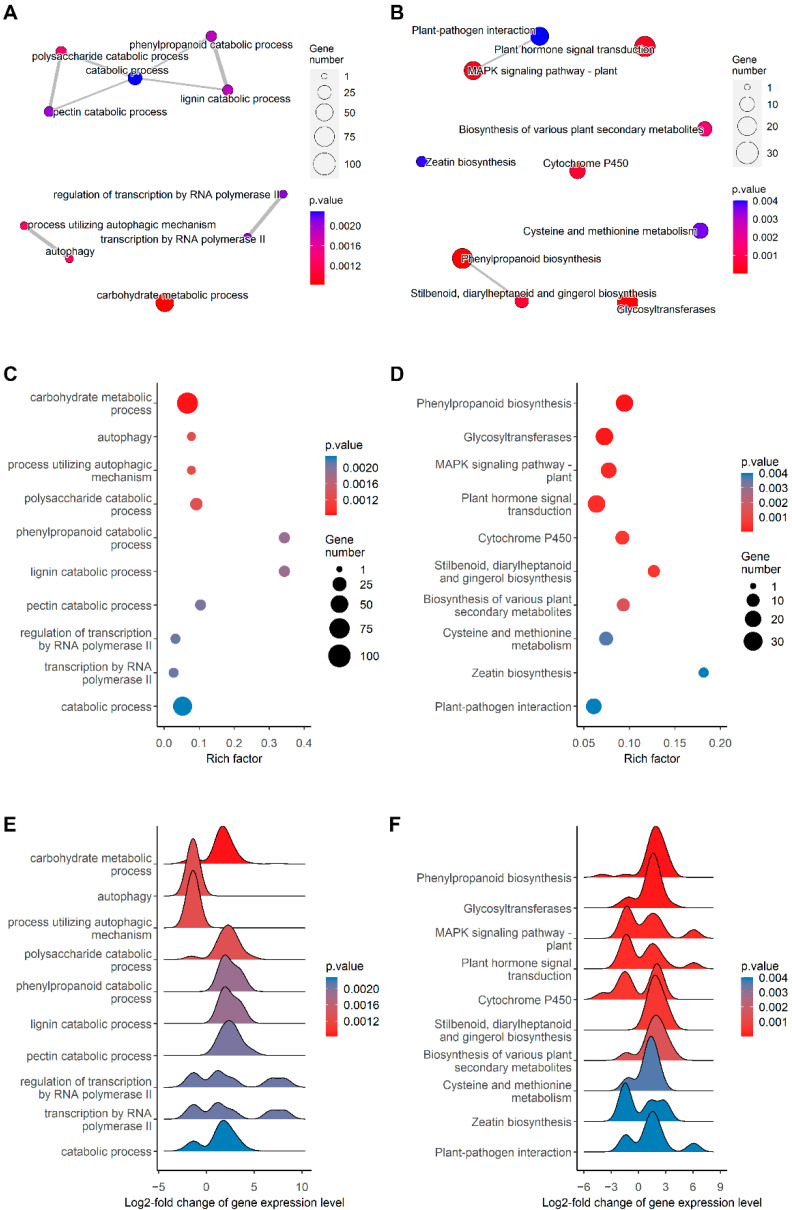
Enrichment analysis of DEGs based on the functional annotation from Gene Ontology and KEGG databases. The DEGs were identified using DESeq2 settings of adjusted *p*-value < 0.1 and absolute log_2_ fold-change > 0.5. The Gene Ontology enrichment analysis was applied using function “gseGO” of the R package “clusterProfiler”, and the KEGG pathway enrichment analysis was applied using function “enricher” of the same package. The plots show the top ten GO terms and KO terms ranked in the order of descending *p*-value. (**A**) Emap plot of Gene Ontology biological processes. Mutually overlapping gene sets tend to cluster together. The dot size indicates the number of genes belonging to the GO term, and the color indicates the significance of the term. (**B**) Emap plot of KEGG pathways. (**C**) Dot plot of the enriched GO terms. The dot size indicates the number of genes belonging to the GO term, and the color indicates the significance of the term. Rich factor is the ratio of the number of DEGs in the term to the total number of the annotated genes in the term. (**D**) Dot plot of the enriched KO terms. (**E**) Ridge plot of the GO terms showing the distribution of the gene expression level of the DEGs in the terms. The gene expression level is measured in log_2_-TPM. (**F**) Ridge plot of the KO terms.

**Figure 6 ijms-23-14746-f006:**
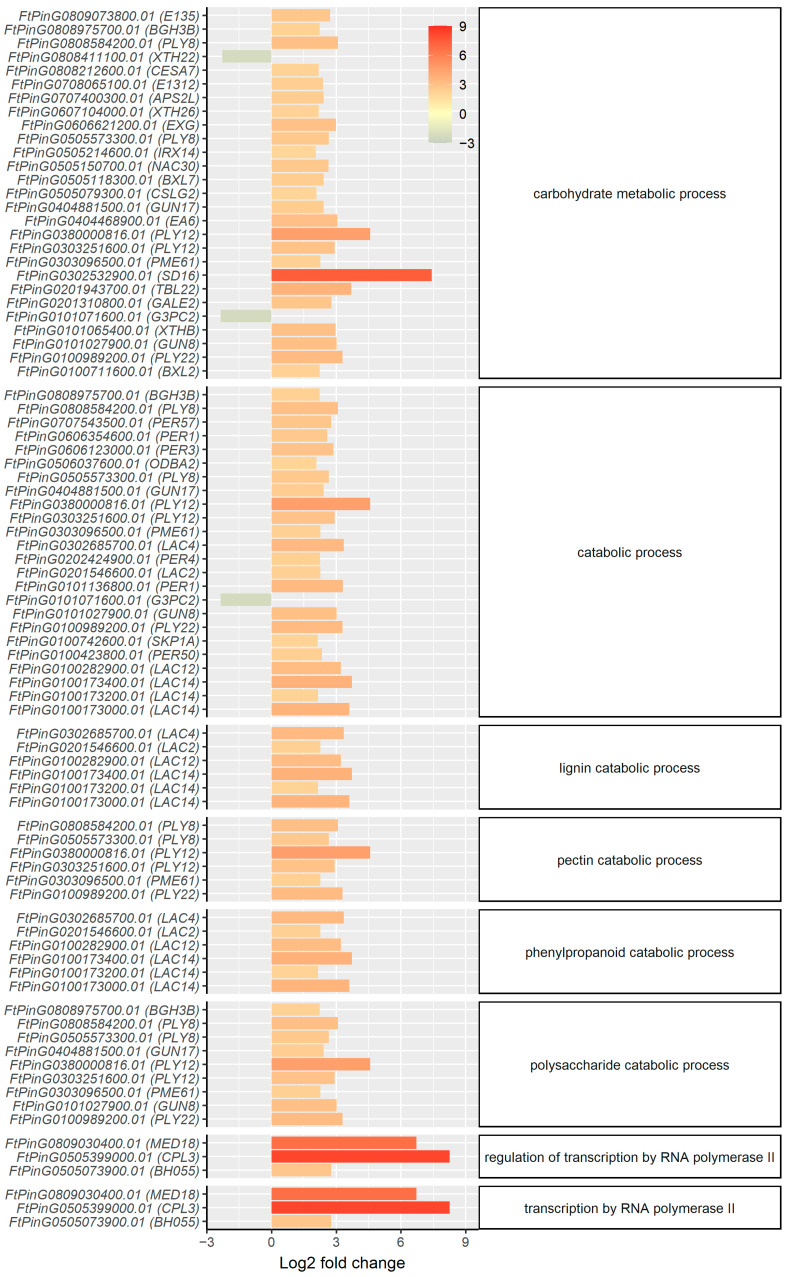
Fold-change of the expression levels of DEGs for the top ten GO terms with lowest *p*-values in the enrichment analysis. The expression level is measured as TPM of the gene. Only genes with an absolute log_2_ fold-change of TPM > 2 are listed in the plot. The labels of the *y* axis denote the gene name, and the annotation of the gene in Swiss-Prot database is shown in brackets.

**Figure 7 ijms-23-14746-f007:**
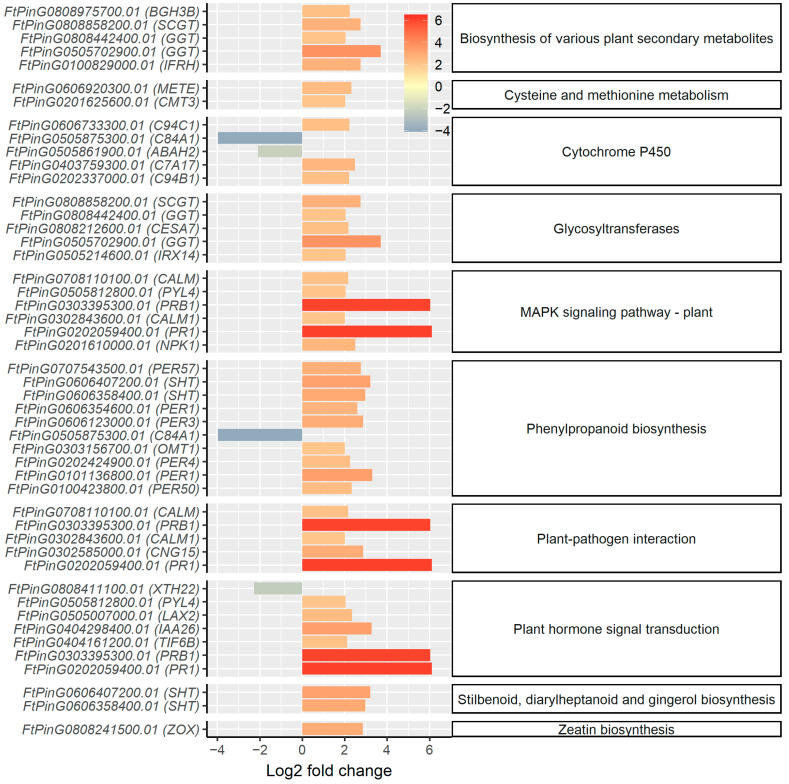
Fold-change of the expression levels of DEGs for the top ten KO terms with lowest *p*-values in the enrichment analysis. The expression level is measured as TPM of the gene. Only genes with an absolute log_2_ fold-change of TPM > 2 are listed in the plot. The labels of the *y* axis denote the gene name, and the annotation of the gene in Swiss-Prot database is shown in brackets.

**Table 1 ijms-23-14746-t001:** H_2_O_2_, MDA, and proline contents, and POD, SOD, and CAT activity levels in the leaves of Tartary buckwheat treated with different concentrations of salicylic acid.

Treatment	H_2_O_2_ Content (µmol g^−1^)	MDA Content (nmol g^−1^)	Pro Content (µg g^−1^)	POD Activity (U g^−1^)	SOD Activity (U g^−1^)	CAT Activity (U g^−1^)
C	0.56 ± 0.01 c	2.5 ± 0.3 c	80 ± 1 ab	768 ± 106 a	631 ± 25 b	393 ± 24 ab
Cd	0.73 ± 0.03 b	5.7 ± 0.1 a	51 ± 2 c	399 ± 22 cd	509 ± 17 c	181 ± 81 c
Cd + SA25	0.88 ± 0.04 a	5.8 ± 0.5 a	67 ± 1 bc	531 ± 10 bc	629 ± 18 b	329 ± 39 abc
Cd + SA50	0.73 ± 0.03 b	5.1 ± 0.2 a	81 ± 6 ab	385 ± 23 d	692 ± 43 ab	270 ± 138 bc
Cd + SA75	0.5 ± 0.05 c	3.6 ± 0.2 b	81 ± 6 ab	492 ± 40 bcd	727 ± 5 ab	373 ± 50 abc
Cd + SA100	0.45 ± 0.03 c	3.2 ± 0.2 bc	98 ± 10 a	601 ± 23 b	797 ± 83 a	526 ± 47 a

C denotes the control plant group cultivated in uncontaminated soil without any treatment. Cd denotes the group of plants cultivated in the soil with 2.0 mg kg^−1^ Cd contamination without SA treatment. Cd + SA25, Cd + SA50, Cd + SA75, and Cd + SA100 denote the plant groups cultivated in soil with 2.0 mg kg^−1^ Cd contamination treated with 25, 50, 75, and 100 µmol L^−1^ salicylic acid, respectively. Different lowercase letters denote significant differences between the treatment groups (*p* < 0.05).

## Data Availability

The data that support the findings of this study are available from the corresponding author upon reasonable request.

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
