# Peer review of "Salicylic Acid Enhances Cadmium Tolerance and Reduces Its Shoot Accumulation in Fagopyrum tataricum Seedlings by Promoting Root Cadmium Retention and Mitigating Oxidative Stress"

_ijms, 2022, doi:10.3390/ijms232314746_

Round 1
Reviewer 1 Report
Manuscript ID: ijms-2050152
Type of manuscript: Article
Title: Salicylic Acid Enhances Cd Tolerance and Reduces Shoot Cd Accumulation in Fagopyrum tataricum Seedlings by Promoting Root Cd Retention and Mitigating Oxidative Stress
Authors: Siwei Luo, Kaiyi Wang, Zhiqiang Li, Hanhan Li, Jirong Shao , Xuemei Zhu
Dear Authors,
First of all I would like to thank the authors for this work. The Abstract is clearly described and comprehensive. The Introduction is also straightforward and clear. The aim of the work was clearly defined and hypothesis was formulated. The authors clearly presented the results and they come to the proper conclusions.
Author Response
Response to Reviewer 1 Comments
We are very grateful for your review of our manuscript, and your positive comments are highly appreciated. We have made minor changes according to the comments of another reviewer, please see the tracked changes of the edited manuscript. In addition, the comments and responses are listed below:
Point 1: Title should be modified and the use of abbrevations should be avoided.
Response 1: We think the abbreviation here refers to element symbol Cd in the title. Although there are plenty of academic publications using chemical element symbols in their titles, we think the point is reasonable and replaced “Cd” with “Cadmium”.
Point 2: Line 11-15: re-write!
Response 2: Line 16-20: We have re-written the two sentences. In order to make the meaning more precise, we replaced “affect the production and quality” to “reduce the production and quality”. In addition, we have reduced the complexity of both sentences.
Point 3: Line 16: show? The results should always be in past tense!!!
Response 3: We also believed that past tense should be used. The verb “show” and the following verb “enhance” were changed to present tense in the English language editing by MDPI. We contacted the English editors again, and the explanation for the point is quoted below:
“’show’ is definitely correct in this case. The reviewer is incorrect that the results must always be expressed in the past tense. When the results are fixed and will not change in the future, the present tense (i.e., "show") should be used.”
“The reviewers are correct in that (in specific circumstances) you would use the past tense to explain your results, e.g., you would not say ‘the root-to-stem translocation factor greatly decreases by 59%’, you would say ‘the root-to-stem translocation factor greatly decreased by 59%’ because it decreased during your experiment, so it needs to be in the past tense. However, when talking about what the results show/indicate/reveal in general, when you are stating an established fact or when the results are fixed and will not change in the future (because you have already carried out your study, so the results cannot change), the present tense is correct.”
So, we would like to keep the present tense.
We have then checked the tense of the verbs throughout the manuscript to ensure their proper usage.
Point 4: Extensive English editing is required for this manuscript to be considered for publication in IJMS.
In my opinion, the authors have explained thier results and conslusions more precisely in the abstract section. Based on the measurements, the could have done much better than the current one.
Response 4: The manuscript has undergone English language editing by MDPI, and we will make every effort to make the manuscript clearer. The following modifications has been made:
Line 94-97: We firstly introduced the influence of cadmium on the plant growth, and then wrote the topic sentence that describes the effect of salicylic acid on the plant under cadmium stress. In this way, the introduction to the results is more straightforward. The readers do not have to read the topic sentence of salicylic acid effect and turn to cadmium influence, and then turn back to salicylic acid effect.
Line 127-130: In the previous version, we described the effect of salicylic acid on the oxidative stress parameters one by one. In the second version, we introduced the effect of 25 µmol L-1 salicylic acid at first, because the effect of salicylic acid at this concentration did not have an obvious ameliorative effect on the oxidative stress.
Point 5: Line 30: The Cd abbrevation was already mentioned in the abstract. This issue should be checked for the other abbrevations throughout the manuscript.
Response 5: All abbreviations were checked throughout the manuscript.
Point 6: Why the authors only used salicylic acid only?
Response 6: According to the previous studies, similar stress responsive plant hormones like abscisic acid and jasmonic acid also have a positive effect on the cadmium stress tolerance in crops, however, we found that salicylic acid is most widely applied in different species in the comparable studies, so we chose salicylic acid. We agree that testing the effect of other plant hormones as well as other plant secondary metabolites should be promising in future studies.
The above-mentioned reason was added to the introduction section (line 48-49).
Point 7: What was the reason of choosing these concentrations of SA?
Response 7: In the comparable studies, the applied concentration of exogenous salicylic acid for crop seedlings are mostly 100 µmol L-1 or under 100 µmol L-1. Therefore, we divided the range evenly to 25, 50, 75 and 100 µmol L-1. The mentioned reason was added to the method section (line 503-505).
Point 8: The figures configuration should be improved especially the y-axis. For instance: C, Cd, Cd+SA25, Cd+SA50, Cd+SA75, Cd+SA100.
Response 8: Your suggestion was greatly appreciated. The axis labels were changed according to the suggestion throughout the manuscript.
Point 9: Why the H2O2, MDA and proline levels were higher in Cd+SA25 than Cd alone?
Response 9: The differences of MDA and proline levels between group Cd and group Cd+SA25 were not statistically significant (p<0.05). The H2O2 content was significantly higher in group Cd+SA25, but SOD activity was also significantly higher (p<0.05), which gave contradictory information of the level of oxidative stress. As all the other four parameters including MDA, proline, POD and CAT did not show significant changes (p<0.05), we tend to believe the higher H2O2 level was caused by chance. In addition, the H2O2 level of group Cd+SA25 had a higher standard deviation than group Cd, and the difference between the groups was not highly significant (p<0.01).
The corresponding section in the manuscript was also improved to make this point clearer (line 120-142).
Point 10: Figure 4D: There should be borders to differentiate the individual boxes.
Response 10: The borders were added.
Point 11: The conclusion section is missing.
Response 11: We added a conclusion section.
Point 12: The prensentation of the experimental should be improved too. In the current version, it is not easy to follow.
Response 12: We have made minor changes to make it clearer (line 488, 494-495, 502-505).

Reviewer 2 Report
Comments
In general, the submitted manuscript to IJMS is set out to assess the ameliorative effects of salicylic acid in fagopyrum tataricum seedlings cadmium-induced oxidative stress. I trust that the paper has the publication potential, but it should be improved in various aspects, as mentioned in the following comments:
Title should be modified and the use of abbrevations should be avoided.
Line 11-15: re-write!
Line 16: show? The results should always be in past tense!!!
Extensive English editing is required for this manuscript to be considered for publication in IJMS.
In my opinion, the authors have explained thier results and conslusions more precisely in the abstract section. Based on the measurements, the could have done much better than the current one.
Line 30: The Cd abbrevation was already mentioned in the abstract. This issue should be checked for the other abbrevations throughout the manuscript.
Why the authors only used salicylic acid only?
What was the reason of choosing these concentrations of SA?
The figures configuration should be improved especially the y-axis. For instance: C, Cd, Cd+SA25, Cd+SA50, Cd+SA75, Cd+SA100.
Why the H2O2, MDA and proline levels were higher in Cd+SA25 than Cd alone?
Figure 4D: There should be borders to differentiate the individual boxes.
The conclusion section is missing.
The prensentation of the experimental should be improved too. In the current version, it is not easy to follow.

Author Response
Response to Reviewer 2 Comments
We are very grateful for your reviews of the manuscript. The comments are encouraging, and we edited our manuscript according to the comments. Please see the detailed response below. The line number mentioned in the response refers to the edited manuscript with tracked changes.
Point 1: Title should be modified and the use of abbrevations should be avoided.
Response 1: We think the abbreviation here refers to element symbol Cd in the title. Although there are plenty of academic publications using chemical element symbols in their titles, we think the point is reasonable and replaced “Cd” with “Cadmium”.
Point 2: Line 11-15: re-write!
Response 2: Line 16-20: We have re-written the two sentences. In order to make the meaning more precise, we replaced “affect the production and quality” to “reduce the production and quality”. In addition, we have reduced the complexity of both sentences.
Point 3: Line 16: show? The results should always be in past tense!!!
Response 3: We also believed that past tense should be used. The verb “show” and the following verb “enhance” were changed to present tense in the English language editing by MDPI. We contacted the English editors again, and the explanation for the point is quoted below:
“’show’ is definitely correct in this case. The reviewer is incorrect that the results must always be expressed in the past tense. When the results are fixed and will not change in the future, the present tense (i.e., "show") should be used.”
“The reviewers are correct in that (in specific circumstances) you would use the past tense to explain your results, e.g., you would not say ‘the root-to-stem translocation factor greatly decreases by 59%’, you would say ‘the root-to-stem translocation factor greatly decreased by 59%’ because it decreased during your experiment, so it needs to be in the past tense. However, when talking about what the results show/indicate/reveal in general, when you are stating an established fact or when the results are fixed and will not change in the future (because you have already carried out your study, so the results cannot change), the present tense is correct.”
So, we would like to keep the present tense.
We have then checked the tense of the verbs throughout the manuscript to ensure their proper usage.
Point 4: Extensive English editing is required for this manuscript to be considered for publication in IJMS.
In my opinion, the authors have explained thier results and conslusions more precisely in the abstract section. Based on the measurements, the could have done much better than the current one.
Response 4: The manuscript has undergone English language editing by MDPI, and we will make every effort to make the manuscript clearer. The following modifications has been made:
Line 94-97: We firstly introduced the influence of cadmium on the plant growth, and then wrote the topic sentence that describes the effect of salicylic acid on the plant under cadmium stress. In this way, the introduction to the results is more straightforward. The readers do not have to read the topic sentence of salicylic acid effect and turn to cadmium influence, and then turn back to salicylic acid effect.
Line 127-130: In the previous version, we described the effect of salicylic acid on the oxidative stress parameters one by one. In the second version, we introduced the effect of 25 µmol L-1 salicylic acid at first, because the effect of salicylic acid at this concentration did not have an obvious ameliorative effect on the oxidative stress.
Point 5: Line 30: The Cd abbrevation was already mentioned in the abstract. This issue should be checked for the other abbrevations throughout the manuscript.
Response 5: All abbreviations were checked throughout the manuscript.
Point 6: Why the authors only used salicylic acid only?
Response 6: According to the previous studies, similar stress responsive plant hormones like abscisic acid and jasmonic acid also have a positive effect on the cadmium stress tolerance in crops, however, we found that salicylic acid is most widely applied in different species in the comparable studies, so we chose salicylic acid. We agree that testing the effect of other plant hormones as well as other plant secondary metabolites should be promising in future studies.
The above-mentioned reason was added to the introduction section (line 48-49).
Point 7: What was the reason of choosing these concentrations of SA?
Response 7: In the comparable studies, the applied concentration of exogenous salicylic acid for crop seedlings are mostly 100 µmol L-1 or under 100 µmol L-1. Therefore, we divided the range evenly to 25, 50, 75 and 100 µmol L-1. The mentioned reason was added to the method section (line 503-505).
Point 8: The figures configuration should be improved especially the y-axis. For instance: C, Cd, Cd+SA25, Cd+SA50, Cd+SA75, Cd+SA100.
Response 8: Your suggestion was greatly appreciated. The axis labels were changed according to the suggestion throughout the manuscript.
Point 9: Why the H2O2, MDA and proline levels were higher in Cd+SA25 than Cd alone?
Response 9: The differences of MDA and proline levels between group Cd and group Cd+SA25 were not statistically significant (p<0.05). The H2O2 content was significantly higher in group Cd+SA25, but SOD activity was also significantly higher (p<0.05), which gave contradictory information of the level of oxidative stress. As all the other four parameters including MDA, proline, POD and CAT did not show significant changes (p<0.05), we tend to believe the higher H2O2 level was caused by chance. In addition, the H2O2 level of group Cd+SA25 had a higher standard deviation than group Cd, and the difference between the groups was not highly significant (p<0.01).
The corresponding section in the manuscript was also improved to make this point clearer (line 120-142).
Point 10: Figure 4D: There should be borders to differentiate the individual boxes.
Response 10: The borders were added.
Point 11: The conclusion section is missing.
Response 11: We added a conclusion section.
Point 12: The prensentation of the experimental should be improved too. In the current version, it is not easy to follow.
Response 12: We have made minor changes to make it clearer (line 488, 494-495, 502-505).
